# Effects of Elevated CO_2_ on the Fitness of Three Successive Generations of *Lipaphis erysimi*

**DOI:** 10.3390/insects13040333

**Published:** 2022-03-29

**Authors:** Xue-Mei Li, Mu-Hua Zhao, Feng Huang, Fang-Ge Shang, Yun-Hui Zhang, Cheng-Min Liu, Shuai-Jie He, Gang Wu

**Affiliations:** 1Hubei Key Laboratory of Insect Resource Utilization and Sustainable Pest Management, College of Plant Science and Technology, Huazhong Agricultural University, Wuhan 430070, China; lxm0616@webmail.hzau.edu.cn (X.-M.L.); zmh@webmail.hzau.edu.cn (M.-H.Z.); s1116@webmail.hzau.edu.cn (F.-G.S.); zhangyh_27@webmail.hzau.edu.cn (Y.-H.Z.); lcm123456@webmail.hzau.edu.cn (C.-M.L.); heshuaijie1015@webmail.hzau.edu.cn (S.-J.H.); 2Eco-Environmental Engineering Evaluation Center, Hubei Academy of Eco-Environmental Sciences, Wuhan 430072, China; huangfeng@hbaes.ac.cn

**Keywords:** elevated CO_2_, *Lipaphis erysimi*, two-sex life table, fecundity, intrinsic rate of increase

## Abstract

**Simple Summary:**

Global warming caused by the increase in atmospheric CO_2_ concentration is becoming a major environmental issue. *Lipaphis erysimi* is one of the most damaging pests of cruciferous crops worldwide, and *L. erysimi* has strong adaptability to the environment and reproductive capacity. The age-stage, two-sex life table is currently used by many researchers in place of the traditional age-specific life table, providing many details such as fitness and potential damage. In this study, the individual fitness and population dynamics parameters of three successive generations of *L. erysimi* were analyzed using the age-stage, two-sex life table. The results show that a high CO_2_ concentration had a cumulative effect on the survival rate and fecundity of *L. erysimi*, and elevated CO_2_ had a negative effect on the individual fitness parameters of *L. erysimi.* The life expectancy (*e_xj_*) is significantly lower in elevated CO_2_ than that in ambient CO_2_ treatment in the three successive generations, indicating that *L. erysimi* was more sensitive to CO_2_ concentration and the life of *L. erysimi* was shortened under elevated CO_2_. Additionally, we can find that elevated CO_2_ has a short-term effect on the population parameters, including the intrinsic rate of increase (*r*) and finite rate of increase (*λ*) in *L. erysimi.* Through the data from this experiment, we believe that the individual and population fitness of *L. erysimi* will be decreased under elevated CO_2_, which indicates that the damage caused by *L. erysimi* may be reduced in the future with increasing CO_2_ levels.

**Abstract:**

To assess the effect of elevated CO_2_ on the development, fecundity, and population dynamic parameters of *L. erysimi*, the age-stage, two-sex life table was used to predict the individual fitness and population parameters of three successive generations of *L. erysimi* in this study. The results show that a significantly longer total pre-adult stage before oviposition (TPOP) was observed in the third generation compared with the first generation of *L. erysimi* under the 800 μL/L CO_2_ treatment. The fecundity is significantly lower in the 800 μL/L CO_2_ treatment than that in the 400 μL/L CO_2_ treatment in the third generation of *L. erysimi*, which indicates that elevated CO_2_ had a negative effect on the individual fitness parameters of *L. erysimi*. Additionally, the life expectancy (*e_xj_*) is significantly lower under the 800 μL/L CO_2_ treatment than that under the 400 μL/L CO_2_ treatment in the three successive generations. A significantly higher intrinsic rate of increase (*r*) and finite rate of increase (*λ*) were found in the second generation compared with those in the first and third generations of *L. erysimi* under the 800 μL/L CO_2_ treatment. Moreover, significantly lower *r* and *λ* were observed under the 800 μL/L CO_2_ treatment compared with those under the 400 μL/L and 600 μL/L CO_2_ treatments in the first generation of *L. erysimi*, which indicates that elevated CO_2_ has a short-term effect on the population parameters (*r* and *λ*) of *L. erysimi*. Our experiment can provide the data for the comprehensive prevention and control of *L. erysimi* in the future with increasing CO_2_ levels.

## 1. Introduction

The atmospheric CO_2_ concentration is increasing year on year due to the use of fossil fuels such as coal, farmland, and factory waste discharge, and the improper utilization of land, alongside human deforestation, especially widespread in tropical forests, causing the concentration of greenhouse gases such as CO_2_ to become higher and higher [1,2]. Global climate change has aroused widespread concern worldwide. NOAA (2021) reported that the atmospheric CO_2_ concentration has risen from 280 μL/L before the Industrial Revolution to 415 μL/L at present, with an annual growth rate of approximately 2.5 ppm in the past five years, and it is expected to reach twice the current level by the end of the 21st century [3].

Since carbon is the key element in the structure of plants, an increased CO_2_ concentration enables faster growth due to rapid carbon assimilation [4]. In general, elevated CO_2_ concentration affects foliar protein [5], leaf biomass [6], water use efficiency (WUE) [7], yield [8,9], and, in turn, the production of carbon (C)-based secondary metabolites [10,11]. After analyzing 122 studies, Robinson et al. [12] concluded that elevated CO_2_ increased the relative consumption rate of arthropods (+14%) but reduced the relative growth rate (−4.5%), as well as the pupal and adult weights (−5.5%). As a vital limiting factor for phytophagous arthropods, the changes in foliar secondary metabolites may have major effects on arthropod performance [13]. Most published documents reported that elevated CO_2_ indirectly influenced arthropod performance via the changes in plant chemical composition [13,14,15,16]. Under the condition of high CO_2_ concentration, the population of *Sitobion avenae* and *Aphis gossypii* increased greatly after feeding on spring wheat [17]. Wen et al. [18] found that elevated CO_2_ can enhance the population parameters of *Nilaparvata lugens*, including the intrinsic rate of increase (*r*), finite rate (*λ*), and net reproductive rate (*R*_0_). Whittaker [19] summarized more than 30 papers and found phloem sap-suckers to have a complex response to elevated CO_2_. For example, there was a phenomenon of positive responses of piercing–sucking insects to elevated CO_2_, such as higher compensation consumption and lower interspecific competition [20]. Newman et al. [21] considered that sucking pests’ responses to elevated CO_2_ are frequently “species-specific”, being negative, positive, or neutral.

*Lipaphis erysimi* (Kaltenbach) is one of the most damaging pests of cruciferous crops worldwide, particularly of vegetable *Brassicae* including cabbage, broccoli, collard, kale, mustard, rape, and turnip [22]. *L. erysimi* has strong adaptability to the environment and reproductive capacity. Adults and nymphs of *L. erysimi* pierce and suck the juice of host plants, leading to the loss of plant nutrients and water imbalance, which reduces the yield of crops and the nutritional value of vegetables. Honeydew secreted by *L. erysimi* can also cause coal stain disease, affecting the quality of products. High aphid densities distort actively growing leaves, causing them to curl, forming pockets and folds that offer shelter to the aphids, thus enabling them to escape insecticide treatments. *L. erysimi* is also an important vector of several different viral diseases [22].

Most published documents focused mainly on the short-term or single-generation responses of *L. erysimi* to CO_2_ enrichment [22,23,24]. However, few of these experiments examined multiple generations of this herbivorous pest with regard to their response to elevated CO_2_. In this study, three successive generations of *L. erysimi* were reared in current/ambient (400 μL/L), medially elevated (600 μL/L) and highly elevated (800 μL/L) levels of CO_2_ concentrations. The developmental time, survival, and fecundity of *L. erysimi* were analyzed using an age-stage, two-sex life table to predict the fitness and potential population damage of *L. erysimi*, which can provide a valuable reference for the effective integrated control of *L. erysimi* with the increasing atmospheric CO_2_ concentrations anticipated in the future.

## 2. Materials and Methods

### 2.1. Closed-Dynamics CO_2_ Chamber

All experiments were performed in controlled environment growth chambers (PRX-450D-30; Haishu Safe Apparatus, Ningbo, China). The growth chambers were maintained at 70 ± 10% RH, 27 ± 1 °C, and a 14L:10D photoperiod with 30,000 LX provided by thirty-nine 26 W fluorescent bulbs. Three levels of CO_2_ concentration were continuously applied, 400 μL/L (the current ambient CO_2_ concentration level), 600 μL/L (medially elevated CO_2_ concentration level, representing the mid-century CO_2_ concentration), and 800 μL/L (double the current ambient CO_2_ concentration level, representing the predicted level by the end of this century), respectively. The growth chambers were equipped with an automatic control system to monitor and adjust the CO_2_ concentration every 20 min, as described in detail in Chen et al. [25].

### 2.2. Host Plants and L. erysimi Stock

Seeds of *Brassica pekinensis*, a cabbage cultivar susceptible to *L. erysimi*, were planted in the field. *L. erysimi* were obtained from greenhouses at the Huazhong Agriculture University and reared more than 50 generations in Hubei Insect Resources Utilization and Sustainable Pest Management Key Laboratory, Wuhan, China. After the seeds sprouted, three-leaf cabbages were put into each closed-dynamics CO_2_ chamber for *L. erysimi* feeding. No chemical fertilizers or insecticides were applied throughout the period of the experiment.

### 2.3. L. erysimi Feeding

Under each CO_2_ concentration level, 60 aphids were observed in each generation. Observing them at 8:00 and 20:00 each day, the ecdysis, and survival of *L. erysimi* were recorded in detail until reaching adulthood. After the emergence of adults, the litter size was observed and recorded every day until all of the test insects died. The ecdysis and newly produced aphids were removed every time. If it was observed that the leaves were yellow and wilting, it was necessary to replace the leaves in time.

### 2.4. Life Table Analysis and Population Projection of L. erysimi

The growth rate, developmental time, and survival of *L. erysimi* were analyzed based on the age-stage, two-sex life table theory. After the emergence of *L. erysimi* adults, the fecundity of *L. erysimi* was calculated by counting the number of nymphs every day. All raw data of life history were analyzed by a computer program, TWOSEX-MSChart [26]. The age-stage-specific survival rate (*s_xj_*) (*x* = age, *j* = stage), the age-specific survival rate (*l_x_*), the female age-stage fecundity (*f* (*x*, female)), the age-specific fecundity (*m_x_*), age-specific maternity (*l_x_*m_x_*), the age-stage life expectancy (*e_xj_*), the age-stage-specific reproductive value (*v_xj_*), and the population parameters (including the net reproductive rate (*R_0_*), the intrinsic rate of increase (*r*), the finite rate of increase (*λ*), and the mean generation time (*T*)) were calculated. The age-stage-specific survival rate (*s_xj_*) means the survival rate of *L. erysimi* at age *x* and stage *j.* The age-specific survival rate (*l_x_*) means the total survival rate at age *x.* The age-stage female fecundity (*f*(*x*, female)) represented the fecundity of females at age *x*, and the age-specific fecundity (*m_x_*) was the fecundity at age *x.* The age-stage life expectancy (*e_xj_*) means that the individual would live until age *x* and stage *j.* The age-stage-specific reproductive value (*v_xj_*) was defined as the contribution of an individual of age *x* and stage *j* to the future population. All of the values above were described in detail in Chi and Liu [27] and Chi [28]. In this study, the means and standard errors of the life table parameters were estimated using the bootstrap technique with 100,000 bootstrap replicates, and the differences among three CO_2_ treatments were compared using the paired bootstrap test at the 5% significance level. The bootstrap technique and the paired bootstrap test are both embedded in the computer program TWOSEX-MSChart. All graphs were created by Sigmaplot 12.0 (Systat Software, SAN Jose, CA, USA).

## 3. Results

### 3.1. Life History Parameters of L. erysimi

Table 1 shows that a significantly longer TPOP of *L. erysimi* was found in the third generation compared with the first generation under the 800 μL/L treatment (*p* < 0.05). However, A significantly lower survival rate of *L. erysimi* was observed in the third generation compared with the first generation under the 800 μL/L treatment (*p* < 0.05). The fecundity of *L. erysimi* increased significantly in the second generation compared with the third generation under the 800 μL/L treatment (*p* < 0.05). A significantly higher fourth instar period of *L. erysimi* was found in the third generation under the 800 μL/L treatment relative to the 400 μL/L and 600 μL/L treatments (*p* < 0.05). Meanwhile, TPOP in the third generation of *L. erysimi* decreased significantly under the 600 μL/L treatment relative to 800 μL/L treatment (*p* < 0.05). In addition, significantly lower adult longevity and survival rate in the third generation of *L. erysimi* were observed for the 800 μL/L treatment relative to the 400 μL/L and 600 μL/L treatments (*p* < 0.05) (Table 1).

Figure 1 shows the age-stage-specific survival rate (*s_xj_*) of three successive generations of *L. erysimi* in response to three CO_2_ concentrations. The results show that the lowest survival rate was found in the third instar of the first generation under the 400 μL/L treatment relative to the 600 μL/L and 800 μL/L CO_2_ treatments (Figure 1). In Figure 2, the largest reproductive contribution (*v_xj_*) (*v_xj_* ≥ 29) of the third generation can be observed under the 600 μL/L CO_2_ treatment during the reproductive peak of *L. erysimi* (Figure 2). The age-specific survival rate (*l_x_*) and age-specific fecundity (*f_xj_*) reflected the growth of *L. erysimi* with age (days), survival, and its fecundity (Figure 3). The results show that the age-specific survival rate (*l_x_*) declined rapidly in the first and third generations under the 800 μL/L CO_2_ treatment, and the survival rate under the 800 μL/L CO_2_ treatment is significantly lower than those for the 400 μL/L and 600 μL/L CO_2_ treatments (Figure 3). Significantly, lower peaks of age-specific fecundity (*m_x_*) were found for the first and third generations under the 800 μL/L treatment compared with the 400 μL/L and 600 μL/L CO_2_ treatments, which indicates that 800 μL/L CO_2_ treatment had an inhibitory effect on the fecundity of *L. erysimi* (Figure 3). Figure 4 shows that the highest age-stage-specific life expectancy (*e_xj_*) of *L. erysimi* was under the 400 μL/L CO_2_ treatment, followed by the 600 μL/L CO_2_ treatment. The lowest age-stage-specific life expectancy (*e_xj_*) of *L. erysimi* was under the 800 μL/L CO_2_ treatment, and the *e_xj_* value is 15.02 (Figure 4). The results indicate that the life span of *L. erysimi* tends to be shortened under the 800 μL/L CO_2_ treatment relative to the other two CO_2_ treatments (Figure 4).

### 3.2. Population Parameters of L. erysimi

Table 2 shows that a significantly higher intrinsic rate of increase (*r*) and finite rate of increase (*λ*) of *L. erysimi* were found in the second generation relative to the first and third generations under the 800 μL/L treatment (*p* < 0.05). The *r* and *λ* values in the third generation of *L. erysimi* under the 800 μL/L treatment were significantly lower than those for the 400 μL/L and 600 μL/L treatments (*p* < 0.05). A significantly higher net reproductive rate (*R_0_*) of *L. erysimi* was found in the second generation relative to the first and third generations under the 800 μL/L treatment (*p* < 0.05). Additionally, the *R_0_* value decreased significantly in the third generation of *L. erysimi* under the 800 μL/L treatment compared with the 400 μL/L and 600 μL/L CO_2_ treatments (*p* < 0.05). A significantly higher mean generation time (*T*) of *L. erysimi* was found in the third generation relative to the second generation under three CO_2_ concentrations (*p* < 0.05). However, there were no differences in *T* value in the second and third generations among three CO_2_ concentrations (*p* < 0.05) (Table 2).

## 4. Discussion

Life table data can provide a comprehensive understanding of the development, survivorship, and fecundity of a population cohort of herbivorous insects, revealing the fitness of a population in variable biotic and abiotic conditions. Compared with the traditional age-specific life table, the two-sex life table incorporates the male component of a population as well as the stage differentiation of population individuals [14]. The present study measured the effect of elevated CO_2_ on the individual life history and population dynamics parameters of three successive generations of *L. erysimi.* The results show that a significantly longer TPOP was observed in the third generation compared with the first generation of *L. erysimi* under the 800 μL/L CO_2_ treatment, which indicates that elevated CO_2_ can lengthen the TPOP duration and lead to the aggravation of the damage caused by *L. erysimi* larvae. The fecundity is significantly lower under the 800 μL/L CO_2_ treatment than under the 400 μL/L CO_2_ treatment in the third generation of *L. erysimi.* The results indicate that a high CO_2_ concentration had a cumulative effect on the fecundity of *L. erysimi*, and elevated CO_2_ had a negative effect on the production of *L. erysimi*. In addition, a significantly lower survival rate was found under the 800 μL/L CO_2_ treatment compared with that under the 400 μL/L CO_2_ treatment in the third generation of *L. erysimi*, which shows that elevated CO_2_ had an adverse effect on the survival rate of *L. erysimi*. The fecundity and survival rate results indicate that elevated CO_2_ has a negative effect on the fitness of *L. erysimi.* The life expectancy (*e_xj_*) was significantly lower under the 800 μL/L CO_2_ treatment than that under the 400 μL/L CO_2_ treatment in the three successive generations, indicating that the life of *L. erysimi* was shortened under elevated CO_2_ conditions. The results show that the *L. erysimi* was more sensitive to a high CO_2_ concentration, and the survival rate was lower in the third generation under the 800 μL/L CO_2_ treatment. Chi and Liu [27] pointed out that neglecting the variable developmental rate and male population may cause errors in calculating demographic parameters, such as the intrinsic rate of increase, net reproductive rate, and the mean generation time [27]. Chi and Liu [27] and Chi [28] developed a two-sex life table to take the stage differentiation and the male population into consideration. Based on the age-stage, two-sex life table [27,28], Chi and Getz (1988) constructed a mass-rearing program for stage-structured populations [29]. Furthermore, mathematical proofs demonstrating the correctness of applying the age-stage, two-sex life table to insect populations were provided by Yu et al. [30] and Chi and Su (2006) [31]. Although aphids develop parthenogenetically during the growing season, the life table of aphids with parthenogenetic development can also be analyzed based on the two-sex life table theory, such as in Seo et al. [32] and Wang et al. [33]. So, the data from the two-sex life table in this present experiment can partially clarify the response of aphids to elevated CO_2_.

The population projections based on the age-stage, two-sex life table are essential for realistic population growth predictions, which provide many details, such as fitness and potential damage. The raw data obtained from the two-sex life table are meaningful compared to the data from the traditional age-specific life table [27]. Because development rates vary widely in a population, stage differentiation is critical to understanding the population ecology of insect herbivores. Thus, the stage structure is essential in projecting the population and the in-population ecology of herbivorous pests at different stages and ages [14]. In the present study, the population parameters, including the intrinsic rate of increase (*r*) and finite rate of increase (*λ*), were significantly higher in the second generation than those in the first and third generations of *L. erysimi* under the 800 μL/L CO_2_ treatment. We can see that the population parameters (such as *r* and *λ*) of *L. erysimi* were different from other population parameters, such as in the mean generation time (*T*) for 800 μL/L CO_2_ between different generations. The reason for this may be that this experiment only examined three generations of *L. erysimi* with regard to their response to elevated CO_2_. Therefore, in future research, we should carry out long-term generations (such as more than ten successive generations) of *L. erysimi* to study their response to elevated CO_2_. Additionally, a significantly higher net reproductive rate (*R_0_*) was found in the second generation compared with those in the first and third generations of *L. erysimi* under the 800 μL/L CO_2_ treatment. However, Fallahpour et al. [34] studied the impact of nitrogen fertilization on the nutritional quality of three canola (*Brassica napus L*.) cultivars (Zarfam, Okapi, and Modena) and on the performance of *L. erysimi*. The net reproductive rate (*R_0_*) in our experiment is higher than that recorded by Fallahpour et al. [34], indicating that there are differences in *R_0_* between our experiment (climate change factor) and this study (host plants and fertilization factors). The results indicate that there are significant differences in the *r* and *λ* of three successive generations of *L. erysimi* among the three CO_2_ treatments. Additionally, a high CO_2_ concentration will have a cumulative effect on the population parameters (*r* and *λ*) of *L. erysimi*, and an 800 μL/L CO_2_ concentration has a more negative effect on the *r* and *λ* of the third generation than on those in the first generation of *L. erysimi*. Moreover, significantly lower *r* and *λ* were observed in insects undergoing the 800 μL/L CO_2_ treatment compared with those undergoing the 400 μL/L and 600 μL/L CO_2_ treatments in the first generation of *L. erysimi*, which indicates that elevated CO_2_ has a short-term instantaneous effect on the population parameters (*r* and *λ*) of *L. erysimi*.

## 5. Conclusions

Our studies provide a profile to exemplify the effect of elevated CO_2_ on the life table, consumption rate, population parameters, and population projection of three successive generations of *L. erysimi* using the age-stage, two-sex life table rather than the traditional age-specific life table. The results show that high CO_2_ concentration had a cumulative effect on the survival rate and fecundity of *L. erysimi*, and elevated CO_2_ had a negative effect on the individual fitness parameters of *L. erysimi.* Combined with the population dynamics parameters, we can conclude that elevated CO_2_ has a short-term instantaneous effect on the population parameters (*r* and *λ*) of *L. erysimi*. Based on the data from this study, we believe that the individual and population fitness of *L. erysimi* will decrease under elevated CO_2_, which can provide a data reference for the comprehensive prevention and control of *L. erysimi* under the future increasing CO_2_ levels.

## Figures and Tables

**Figure 1 insects-13-00333-f001:**
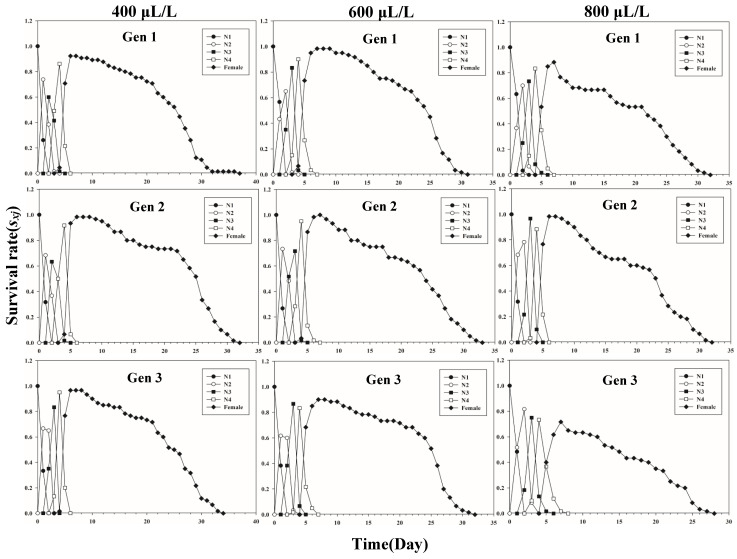
Age-stage-specific survival rates (*s_xj_*) of three successive generations of *L. erysimi* under three CO_2_ treatments.

**Figure 2 insects-13-00333-f002:**
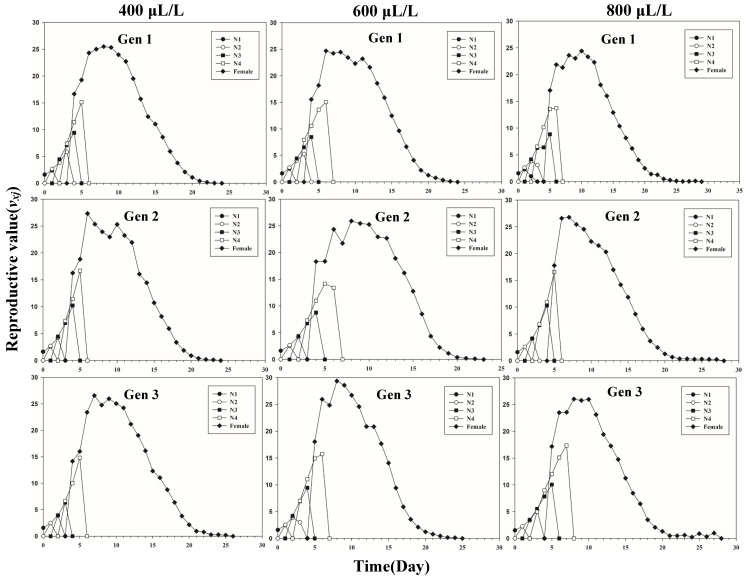
Age-stage-specific reproductive value (*v_xj_*) of three successive generations of *L. erysimi* under three CO_2_ treatments.

**Figure 3 insects-13-00333-f003:**
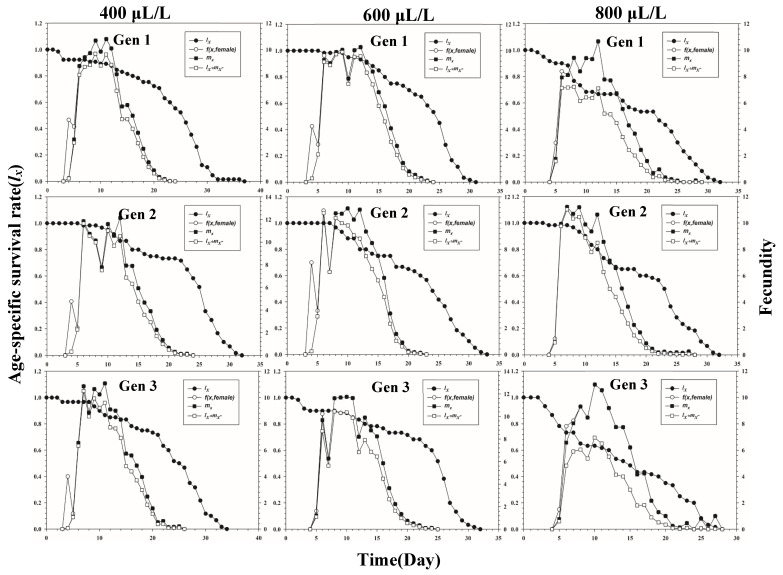
Age-specific survival rate (*l_x_*) (left *y*-axis), age-stage-specific fecundity (*f*(*x*, female)) (right *y*-axis), age-specific fecundity (*m_x_*) (right *y*-axis), and age-specific maternity (*l_x_m_x_*) (right *y*-axis) of three successive generations of *L. erysimi* under three CO_2_ treatments.

**Figure 4 insects-13-00333-f004:**
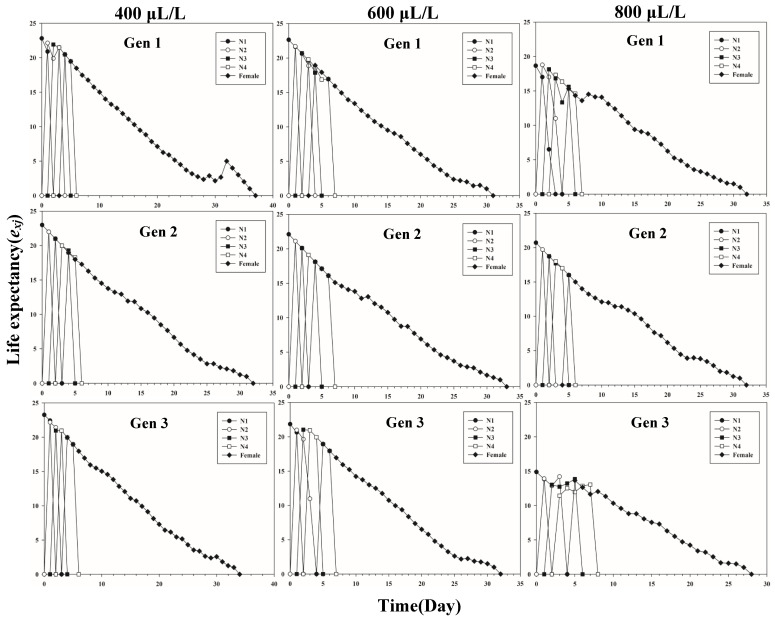
Age-stage-specific life expectancy (*e_xj_*) of three successive generations of *L. erysimi* under three CO_2_ treatments.

**Table 1 insects-13-00333-t001:** Life history parameters of three successive generations of *L. erysimi* under three CO_2_ treatments (mean ± SE).

Life History Parameters	400 μL/L			600 μL/L			800 μL/L		
	Gen 1	Gen 2	Gen 3	Gen 1	Gen 2	Gen 3	Gen 1	Gen 2	Gen 3
1st instar (d)	1.26 ± 0.05 bA	1.32 ± 0.06 aA	1.33 ± 0.06 aA	1.57 ± 0.06 aA	1.27 ± 0.06 aB	1.38 ± 0.06 aB	1.64 ± 0.06 aA	1.32 ± 0.06 aB	1.48 ± 0.06 aAB
2nd instar (d)	1.14 ± 0.04 aB	1.05 ± 0.03 cB	1.32 ± 0.06 abA	1.10 ± 0.04 aA	1.22 ± 0.05 bA	1.20 ± 0.05 bA	1.14 ± 0.05 aB	1.47 ± 0.06 aA	1.45 ± 0.07 aA
3rd instar (d)	1.10 ± 0.04 aA	1.15 ± 0.05 aA	1.21 ± 0.05 bA	1.22 ± 0.05 aB	1.25 ± 0.06 aB	1.44 ± 0.07 aA	1.13 ± 0.05 aB	1.29 ± 0.06 aA	1.17 ± 0.05 bAB
4th instar (d)	1.70 ± 0.06 aA	1.48 ± 0.06 aB	1.33 ± 0.06 bB	1.34 ± 0.06 bA	1.38 ± 0.06 aA	1.24 ± 0.06 bA	1.54 ± 0.07 aA	1.15 ± 0.05 bB	1.57 ± 0.08 aA
APOP (d)	0.20 ± 0.05 aB	0.37 ± 0.06 aA	0.43 ± 0.06 aA	0.14 ± 0.04 aAB	0.05 ± 0.03 bB	0.28 ± 0.06 aA	0.07 ± 0.03 bB	0.42 ± 0.06 aA	0.34 ± 0.07 aA
TPOP (d)	5.38 ± 0.07 aB	5.37 ± 0.08 bB	5.62 ± 0.07 abA	5.36 ± 0.09 aA	5.14 ± 0.06 cB	5.57 ± 0.08 bA	5.52 ± 0.09 aB	5.64 ± 0.06 aAB	5.90 ± 0.13 aA
Fecundity	100.6 ± 3.5 aA	101.4 ± 3.6 aA	100.6 ± 3.8 aA	102.0 ± 2.6 aA	96.5 ± 4.4 aA	106.2 ± 4.0 aA	83.2 ± 5.5 bAB	94.3 ± 4.1 aA	79.8 ± 5.6 bB
Adult longevity (d)	19.3 ± 0.8 aA	18.0 ± 0.9 aA	18.8 ± 0.9 aA	17.7 ± 0.7 aA	17.0 ± 0.9 aA	18.7 ± 0.8 aA	14.9 ± 1.1 bAB	15.8 ± 0.9 aA	13.0 ± 1.0 bB
Survival rate (%)	0.92 ± 0.03 abB	1.00 ± 0.00 aA	0.97 ± 0.02 aAB	0.98 ± 0.02 aA	1.00 ± 0.00 aA	0.90 ± 0.04 aB	0.90 ± 0.04 bB	0.98 ± 0.02 aA	0.73 ± 0.06 bC

Note: Standard errors were analyzed by using 100,000 bootstraps. Means followed by lowercase letters indicate significant differences among three CO_2_ treatments within the same generation (*p* < 0.05). Means followed by different uppercase letters indicate significant differences among three generations within the same CO_2_ treatment (*p* < 0.05). APOP: the pre-oviposition period based on the adult stage. TPOP: the total pre-adult stage before oviposition.

**Table 2 insects-13-00333-t002:** Population parameters of successive three generations of *L. erysimi* under three CO_2_ treatments (mean ± SE).

Generations		Population Parameters			
		Intrinsic Rate of Increase (*r*) (d^−1^)	Finite Rate of Increase (*λ*)	Net Reproductive Rate (*R*_0_) (Offspring)	Mean Generation Time (*T*) (d)
400 μL/L	Gen 1	0.4744 ± 0.0076 aA	1.6071 ± 0.0123 aA	92.80 ± 4.63 aA	9.55 ± 0.11 aB
	Gen 2	0.4926 ± 0.0054 aA	1.6365 ± 0.0088 aA	101.42 ± 3.59 aA	9.38 ± 0.10 aB
	Gen 3	0.4552 ± 0.0052 aB	1.5766 ± 0.0081 aB	97.28 ± 4.37 aA	10.06 ± 0.10 aA
600 μL/L	Gen 1	0.4763 ± 0.0057 aA	1.6102 ± 0.0092 aA	100.27 ± 3.07 aA	9.67 ± 0.11 aB
	Gen 2	0.4832 ± 0.0047 abA	1.6213 ± 0.0076 abA	96.52 ± 4.43 aA	9.46 ± 0.10 aB
	Gen 3	0.4573 ± 0.0067 aB	1.5799 ± 0.0105 aB	95.58 ± 5.44 aA	9.97 ± 0.08 aA
800 μL/L	Gen 1	0.4395 ± 0.0085 bB	1.5519 ± 0.0131 bB	75.06 ± 5.91 bB	9.81 ± 0.12 aAB
	Gen 2	0.4742 ± 0.0039 bA	1.6068 ± 0.0063 bA	92.70 ± 4.30 aA	9.55 ± 0.09 aB
	Gen 3	0.4055 ± 0.0124 bC	1.500 ± 0.0185 bC	58.53 ± 6.10 bB	10.02 ± 0.14 aA

Note: Means followed by lowercase letters indicate significant differences among three CO_2_ treatments within the same generation (*p* < 0.05). Means followed by different uppercase letters indicate significant differences among three generations within the same CO_2_ treatment (*p* < 0.05).

## Data Availability

Data are contained within the article.

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
