# Peer review of "Effects of Elevated CO2 on the Fitness of Three Successive Generations of Lipaphis erysimi"

_insects, 2022, doi:10.3390/insects13040333_

Round 1

Reviewer 1 Report

Climate change will have a big impact on agricultural ecosystems. An increase in CO2 can affect plant physiology directly and indirectly on plant-insect interaction. There is ample evidence that increased levels of CO2 can contribute to greater pest damage to plants. Some aphids are successful in elevated CO2 conditions. It is more likely that aphids are able to overcome the disadvantages of the indirect effects of elevated CO2 by reducing developmental times and increasing fecundity under elevated CO2 conditions.

Therefore, I believe that the research subject of the work presented for reviev is current and valuable. The results of the study may have an application character, especially in the recommended integrated method of plant protection.

My opinion on the manuscript submitted for reviev.

  1. Lipaphis erysimi is a European species. The optimal temperature range for this species is 20-250 So why did the authors conduct the experiment at such a high temperature of 27oC? Did they consider that such a high temperature could have been a stress factor for the aphid and affected demographic parameters? I believe that the authors should have conducted the experiment at a lower temperature, closer to the optimum for this species, or at several temperatures in the 20-28oC range.
  2. Population parameters of erysimi were analyzed by the authors based on the age-stage, two-sex life table theory. My doubt is raised by the very high rates, especially Net reproductive rate Ro. I suggest to compare the research results with other works, such as the paper cited in the manuscript:

Qayyum A., et all. 2018. Demographic Parameters of Lipaphis erysimi (Hemiptera: Aphididae) on Different Cultivars of Brassica Vegetables. J. Econ. Entomol. 2018 or

Fallahpour, F.; Ghorbani, R.; Mahallati, M.N.; Hosseini, M. Demographic parameters of Lipaphis erysimi on canola cultivars under different nitrogen fertilization regimes. J. Agric. Sci. Technol. 2015, 17, 35–47.

In most afidological studies the demographic data are determined according to the methods described by Birch in 1948, and the intrinsic growth rate (rm) calculated with Watt and White formula (1977). I suggest that the data obtained be recalculated using this method and the results confronted.

  1. The figures are in very poor quality and are difficult to interpret. However, they contain numerous errors.

- Fig. 1 - the authors give Gen. 1 twice,

- Fig. 2 and 3 are the same figure, the correct figure 3 is missing.

Please explain why values for nymphs appear in Figure 2. Doesn't reproductive value only apply to adults?

  1. The authors wrote that: „The results indicated that the life span of erysimi tends to be shortened in 800 μL/L CO2 treatment relative to other two CO2 treatments (Fig. 4)”. In my opinion, the period is slightly shorter only for the 3rd generation.
  2. Discussion

I think that the authors have insufficiently interpreted and confronted the results of their study with other authors. For example, can the authors explain or interpret why population parameters of L. erysimi, including intrinsic rate of increase (r) and finite rate of increase (λ) are significantly higher in the 2nd generation than those in the 1st and 3rd generations of L. erysimi in 800 μL/L CO2 treatment?

I also believe that no clear conclusions can be drawn from the results obtained. In my opinion, the observed differences between generations are not spectacular.

  1. I think the paper is carelessly prepared and needs careful proofreading again. I give some examples of errors:

- verse 39 and 258 - missing the letter g by generation,

- Verse 128 - N. lugens? What species did the authors write about?

- Verse 149 - significantly should be written with a lowercase letter,

- Verse 154-155 - the word relative to 800 μL/L is missing

- line 272 - in the future - double occurs

- line 224 - double dot,

- line 135 - did the authors want to add another citation?

- line 172 -175: please reread this sentence: Significantly lower age-specific fecundities (mx) were found in the 1st and 3rd generations in 600 μL/L treatment compared with 400 μL/L and 600 μL/L CO2 treatments, which indicated that 600 μL/L CO2 treatment had the inhibitory effect on reproduction of L. erysimi (Fig. 3). I recall that the authors did not include this figure.

Author Response

Responses to reviewer’s comments

Reviewer #1

Q1:.Climate change will have a big impact on agricultural ecosystems. An increase in CO2 can affect plant physiology directly and indirectly on plant-insect interaction. There is ample evidence that increased levels of CO2 can contribute to greater pest damage to plants. Some aphids are successful in elevated CO2 conditions. It is more likely that aphids are able to overcome the disadvantages of the indirect effects of elevated CO2 by reducing developmental times and increasing fecundity under elevated CO2 conditions.

Therefore, I believe that the research subject of the work presented for reviev is current and valuable. The results of the study may have an application character, especially in the recommended integrated method of plant protection.

Responses: Thank you very much for giving us the opportunity to revise our manuscript. In the revised manuscript, we have revised our manuscript according to your and reviewers’ opinions. I look forward to publishing our manuscript on Insects! And we have lined out the right revised sentence by red words in the revised manuscript. Please check it.

Q2: Lipaphis erysimi is a European species. The optimal temperature range for this species is 20-250 So why did the authors conduct the experiment at such a high temperature of 27oC? Did they consider that such a high temperature could have been a stress factor for the aphid and affected demographic parameters? I believe that the authors should have conducted the experiment at a lower temperature, closer to the optimum for this species, or at several temperatures in the 20-28oC range.

Responses: Thank the reviewer for your useful advice! In this study, we mainly study the impact of global warming, especially the elevated CO2 on the fitness of L. erysimi. In addition, IPCC (2013) predict the CO2 concentration will reach to 809 μL/L in 2090. And the temperature will be more than 2℃ in the end of this century relative to the early of this century. Thus, we set up the temperature by 27℃ (represented the temperature in the end of this century) in this study.

Q3: Population parameters of erysimi were analyzed by the authors based on the age-stage, two-sex life table theory. My doubt is raised by the very high rates, especially Net reproductive rate Ro. I suggest to compare the research results with other works, such as the paper cited in the manuscript:

Qayyum A., et all. 2018. Demographic Parameters of Lipaphis erysimi (Hemiptera: Aphididae) on Different Cultivars of Brassica Vegetables. J. Econ. Entomol. 2018 or Fallahpour, F.; Ghorbani, R.; Mahallati, M.N.; Hosseini, M. Demographic parameters of Lipaphis erysimi on canola cultivars under different nitrogen fertilization regimes. J. Agric. Sci. Technol. 2015, 17, 35–47.

Responses: Thank you very much for your reminder. Thank the reviewer for your useful advice and the valuable documents! Indeed, Net reproductive rate (Ro) is higher in this experiment. Our experimental record showed that the reproductive amount of single female was range from 60 to 100. And the Net reproductive rate (Ro) were relatively moderate (Ro range from 6 to 30) in the experiments by Qayyum A., et al. (2018) and Fallahpour et al. (2015). Fallahpour et al. (2015) carried out the impact of nitrogen fertilizations on nutritional quality of three canola (Brassica napus L.) cultivars (Zarfam, Okapi and Modena), on the performance of L.erysimi (host plants and fertilization factors). And our experiment carried out the effect elevated CO2 (environmental factors) on the fitness of L.erysimi. Thus, there are the difference among our experiment and these two literatures. Please chect in the line 268-274 by red words in page 16. Thanks again to the reviewers for his/her in-depth comments and valuable comments.

Q4:In most afidological studies the demographic data are determined according to the methods described by Birch in 1948, and the intrinsic growth rate (rm) calculated with Watt and White formula (1977). I suggest that the data obtained be recalculated using this method and the results confronted.

Responses: Thank you very much for your reminder. Thank the reviewer for your useful advice and the valuable documents! And the life table data obtained from “Birch in 1948, and Watt and White formula (1977)” can provide researchers with a comprehensive understanding of the development, survivorship and fecundity of a population cohort, this not only reveals the fitness of a population in variable biotic and abiotic conditions, but has increasingly been used as an invaluable tool in successful biological control programs. The age-stage, the two-sex life table was used in this study, primarily because of its ability to incorporate the male component of a population as well as the stage differentiation of population individuals. Female-only life table, by definition, ignore the male, which are normally 50% of a population and do contribute significantly to its ecology. The raw data obtained from a age-stage, tow-sex life table is much more credible and meaningful compared to that obtained from tradition age-specific life table. Because development rates do vary widely in a population, stage differentiation is critical to understanding the population ecology of insects. Thus, The age-stage, the two-sex life table is currently used in this study. Thanks again to the reviewers for his/her valuable comments.

Q5: The figures are in very poor quality and are difficult to interpret. However, they contain numerous errors.

- Fig. 1 - the authors give Gen. 1 twice,

- Fig. 2 and 3 are the same figure, the correct figure 3 is missing.

Responses: Thank you very much for your reminder. In the revised manuscript, Figures 1 and 3 were modified according to your advice. Please check it in Fig. 1 of page 8, and Fig. 2 of page 10 in the revised manuscript!

Q6: Please explain why values for nymphs appear in Figure 2. Doesn't reproductive value only apply to adults?

Responses: Thank the reviewer for your useful advice! Indeed, the “reproductive value was only apply to adults”. In this study, the values for nymphs were appeared in Table 1 and Figure 2. And the data showed that significantly longer TPOP (the total pre-adult stage before oviposition) of L. erysimi was found in 800 μL/L CO2 treatment compared with 400 μL/L CO2 treatment in three successive generations of L. erysimi, which indicated 800 μL/L CO2 treatment is unfavorable for its growth and population dynamics. Thus, the values for nymphs were contributed to Age-stage-specifc reproductive value (vxj) . Thanks again for the reviewer for the valuable comments.

Q7: The authors wrote that: „The results indicated that the life span of erysimi tends to be shortened in 800 μL/L CO2 treatment relative to other two CO2 treatments (Fig. 4)”. In my opinion, the period is slightly shorter only for the 3rd generation.

Responses: Thank the reviewer for your useful advice! We have adjusted the ordinate to a uniform scale. Please check the Fig.4 in the page 14 in the revised manuscript!

Q8:Discussion:I think that the authors have insufficiently interpreted and confronted the results of their study with other authors. For example, can the authors explain or interpret why population parameters of L. erysimi, including intrinsic rate of increase (r) and finite rate of increase (λ) are significantly higher in the 2nd generation than those in the 1st and 3rd generations of L. erysimi in 800 μL/L CO2 treatment?I also believe that no clear conclusions can be drawn from the results obtained. In my opinion, the observed differences between generations are not spectacular.

Responses: Thank the reviewer for your useful advice! Indeed, That's true over the three generations we've looked at, but we're going to look at multiple generations to see what happens.

Q9:I think the paper is carelessly prepared and needs careful proofreading again. I give some examples of errors:

- verse 39 and 258 - missing the letter g by generation,

- Verse 128 - N. lugens? What species did the authors write about?

- Verse 149 - significantly should be written with a lowercase letter,

- Verse 154-155 - the word relative to 800 μL/L is missing

- line 272 - in the future - double occurs

- line 224 - double dot,

- line 135 - did the authors want to add another citation?

- line 172 -175: please reread this sentence: Significantly lower age-specific fecundities (mx) were found in the 1st and 3rd generations in 600 μL/L treatment compared with 400 μL/L and 600μL/L CO2 treatments, which indicated that 600 μL/L CO2 treatment had the inhibitory effect on reproduction of L. erysimi (Fig. 3). I recall that the authors did not include this figure.

Responses: Thank you very much for your reminder. The above problems have been modified. Please check in the article.

Reviewer 2 Report

The manuscript entitled "Effects of elevated CO2 on the fitness of three successive generations of Lipaphis erysimi" brings interesting data on aphid life history under elevated CO2 concentration. Such relatively long-term experiments with observation of three successive generations are needed and it is good to have these data published. Nevertheless, I found several issues which should be corrected or answered prior acceptance.

  • The ecdysis was monitored only twice a day (l. 114). This is quite long interval. Given the length of instar development is usually 1-2 days (Table 1). Given that I suggest omit analyses of particular instar durations and APOP and focus on either total time of development (egg to adult) or TPOP. This would reduce the data, greatly clarify the Results (and whole manuscript). Note also that analysis of particular instars did not bring any interesting results. The particular instar development should be analysed e.g. in the terms of developmental rate isomorphy (see Boukal DS, Ditrich T, Kutcherov D, Sroka P, Dudová P, Papáček M (2015) Analyses of Developmental Rate Isomorphy in Ectotherms: Introducing the Dirichlet Regression. PLoS ONE 10(6): e0129341. https://doi.org/10.1371/journal.pone.0129341).

  • Focusing on a single developmental period (e.g. TPOP), it would allow to use some parametric statistical tests (e.g. repeated –measures ANOVA; with TPOP recorded repeatedly (in every generation) for each CO2 level). Your statistical method is actually not good for your data, as a length of successive instars are not independent on each other.

  • The quality of the figures is low and I did not understand them. I did not get the legend, there are lines for particular instars? Again I suggest information for particular instars, data for total survival (Figure 1) and adults (Figure 2) are sufficient. I am totally confused from the Figure 3 – there should be different left and right y-axes, but I do not see them – you probably inserted Figure 2 again. Also check the Figures titles, some letters are missing (“specifc”; “ftted” etc.)

  • One reference on l. 135 is missing

  • Check the English, there are some grammatical and stylistic errors throughout the text.

Author Response

Reviewer #2

Q1:.The manuscript entitled "Effects of elevated CO2 on the fitness of three successive generations of Lipaphis erysimi" brings interesting data on aphid life history under elevated CO2 concentration. Such relatively long-term experiments with observation of three successive generations are needed and it is good to have these data published. Nevertheless, I found several issues which should be corrected or answered prior acceptance.

Responses: Thank you very much for giving us the opportunity to revise our manuscript. In the revised manuscript, we have revised our manuscript according to your and reviewers’ opinions. I look forward to publishing our manuscript on Insects! And we have lined out the right revised sentence by red words in the revised manuscript. Please check it.

Q2:The ecdysis was monitored only twice a day (l. 114). This is quite long interval. Given the length of instar development is usually 1-2 days (Table 1). Given that I suggest omit analyses of particular instar durations and APOP and focus on either total time of development (egg to adult) or TPOP. This would reduce the data, greatly clarify the Results (and whole manuscript). Note also that analysis of particular instars did not bring any interesting results. The particular instar development should be analysed e.g. in the terms of developmental rate isomorphy (see Boukal DS, Ditrich T, Kutcherov D, Sroka P, Dudová P, Papáček M (2015) Analyses of Developmental Rate Isomorphy in Ectotherms: Introducing the Dirichlet Regression. PLoS ONE 10(6): e0129341. https://doi.org/10.1371/journal.pone.0129341).

 Responses: Thank you very much for your reminder. The experiment is based on the current observation, and we will pay attention to increase manpower to shorten the observation time in the future. We have deleted the APOP and instars analysis in accordance with your advice. Please check it.

Q3:Focusing on a single developmental period (e.g. TPOP), it would allow to use some parametric statistical tests (e.g. repeated –measures ANOVA; with TPOP recorded repeatedly (in every generation) for each CO2 level). Your statistical method is actually not good for your data, as a length of successive instars are not independent on each other.

Responses: Thank you very much for your reminder. Thank the reviewer for your useful advice! Indeed, the “length of successive instars are not independent on each other.” And the definition of each nymph instar period of L. erysimi should be shortened the observation time in this experiment. Thus, our data should be focused on the TPOP according to the reviewere’s advice. In the Results Part of this manuscript, we deleted the results of each instar period (such as 1st instar, 2nd instar, 3rd instar, 4th instar) of L. erysimi. In addition, we should shorten the observation time according to the reviewer’s suggestion, such as observing the nymph instar period every two hours in the future experiment(replacing the nymph instar period every twelve hours in the present study). Thanks again for the reviewers’ for the valuable comments.

Q4:The quality of the figures is low and I did not understand them. I did not get the legend, there are lines for particular instars? Again I suggest information for particular instars, data for total survival (Figure 1) and adults (Figure 2) are sufficient. I am totally confused from the Figure 3 – there should be different left and right y-axes, but I do not see them – you probably inserted Figure 2 again. Also check the Figures titles, some letters are missing (“specifc”; “ftted” etc.)

Responses: Thank you very much for your reminder. Figures 3 have been modified. The words in the title part of the figure have also been modified. Please check it in the line 190, 194, 198,199 and 203 in the revised manuscript!

Q5:One reference on l. 135 is missing

 Responses: Thank you very much for your reminder. This issue has been modified. Please check it in the line 137 in page 3 in the revised manuscript!

Q6:Check the English, there are some grammatical and stylistic errors throughout the text.

Responses: Thank you very much for your reminder. We have tried our best to revise our manuscript according to the reviews’ comments letter by letter, and the amendments are highlighted in red in the revised manuscript.

Reviewer 3 Report

In the manuscript "Effects of elevated CO2 on the fitness of three successive generations of Lipaphis erysimi" investigate the role of high CO2 on the fitness traits in the three generations of aphids. The paper is nicely written, easy to read, stands a curious question, and presents clear results. However, this species of aphids rarely has a sexual generation, therefore, it was not quite clear to me why the authors used the two-sex life table (if they did not measure male characteristics). Perhaps, a more detailed description of the method would help a reader to understand this idea. 

Author Response

Reviewer #3

Q1:In the manuscript "Effects of elevated CO2 on the fitness of three successive generations of Lipaphis erysimi" investigate the role of high CO2 on the fitness traits in the three generations of aphids. The paper is nicely written, easy to read, stands a curious question, and presents clear results. However, this species of aphids rarely has a sexual generation, therefore, it was not quite clear to me why the authors used the two-sex life table (if they did not measure male characteristics). Perhaps, a more detailed description of the method would help a reader to understand this idea. 

Responses: Thank you very much for giving us the opportunity to revise our manuscript. One of advantages of two-sex life table is that it takes into male population compared with traditional life table. While the stage differentiation and the variable developmental rate among individuals is ignored in traditional life table. For most insects, the developmental rate is different between the individuals and the sex. Chi & Liu (1985) pointed out that neglecting the variable developmental rate and male population may cause errors in calculating demographic parameters, such as the intrinsic rate of increase, net reproductive rate and the mean generation time. Chi and Liu (1985) and Chi (1988) developed an two-sex life table to take the stage differentiation and the male population into consideration. Based on the age-stage, two-sex life table, Chi & Getz (1988) constructed a mass-rearing program for stage-structured populations. Furthermore, mathematical proofs demonstrating the correctness of applying the age-stage, two-sex life table to insect populations were provided by Yu et al. (2005) and Chi and Su (2006).

Although aphid develop parthenogenetically during the growing season, life table of aphid with parthenogenetic can also be analyzed based on two-sex life table theory, such as Seo et al. (2020) and Wang et al. (2019).

Reviewer 4 Report

The MS presents the results of quite important experiment on the influence of elevated CO2 concentration on some biological aspects of L. erysimi. From agricultural point of view these may be quite important results and data, showing influence of hich CO2 level on pest aphid fitness. Experiment seems to be well designed and presented, some small linguistic errors might be corrected. 

I would have suggestion concerning the presentation of the results: from my point of view as an aphidologist, it would be more comprehensive to me if tables or figures presented merits on all three generations separately for each CO2 level. In combined data as in Tables 1 and 2 is difficult to find a pattern related to particular CO2 level. 

I also have question: doest fecundity means the number of layed eggs, if oviposition is taken into account and both sexes? For me oviposition refers only to number of eggs layed by oviparous female alfter copulation. This needs to explained in methodological part of the MS. 

I would also not be so sure to conclude that high CO2 concentrations will lead to lower fitness of aphids. You have studied recently evolved aphids under suddenly elevated CO2. In reality, this is going to be a process taking decades during which a whole spectre of ecological factors will influence evolution of hundreds of generations of L. erysimi worldwide leading to occurence of most highly adapted lineages. And it may not result in occurence of lower number of these pests or their lower harmfullness.  

Author Response

Reviewer #4

Q1:The MS presents the results of quite important experiment on the influence of elevated CO2 concentration on some biological aspects of L. erysimi. From agricultural point of view these may be quite important results and data, showing influence of hich CO2 level on pest aphid fitness. Experiment seems to be well designed and presented, some small linguistic errors might be corrected. 

Responses: Thank you very much for your useful suggestions. In the revised manuscript, we have revised our manuscript according to your opinions. And we have lined out the right revised sentence by red words in the revised manuscript. Please check it.

Q2:I would have suggestion concerning the presentation of the results: from my point of view as an aphidologist, it would be more comprehensive to me if tables or figures presented merits on all three generations separately for each CO2 level. In combined data as in Tables 1 and 2 is difficult to find a pattern related to particular CO2 level. 

Responses: Thank you very much for your reminder. We made adjustments to the form. Please check it in the Table 1 of page 5 and Table 2 of page 11.

Q3:I also have question: doest fecundity means the number of layed eggs, if oviposition is taken into account and both sexes? For me oviposition refers only to number of eggs layed by oviparous female alfter copulation. This needs to explained in methodological part of the MS.

Responses: Thank you very much for your reminder. In this experiment, L. erysimi is parthenogenetic, and fecundity refers to the number of offspring produced by the female.

Q4:I would also not be so sure to conclude that high CO2 concentrations will lead to lower fitness of aphids. You have studied recently evolved aphids under suddenly elevated CO2. In reality, this is going to be a process taking decades during which a whole spectre of ecological factors will influence evolution of hundreds of generations of L. erysimi worldwide leading to occurence of most highly adapted lineages. And it may not result in occurence of lower number of these pests or their lower harmfullness.

Responses: Thank you very much for your reminder. I quiet agree with the reviewer’s suggestion. Indeed, CO2 concentration rise is a gradual and slow process. And the effect of CO2 concentration on insects is a long and long-term process. In our future experiment, we should conduct the effect of gradual CO2 concentrations on the multi-generation of L. erysimi.

Round 2

Reviewer 1 Report

Thanks for the authors' answers. Some I can accept, but not all. The authors also did not follow all of my recommendations. I believe that the work sent for re-evaluation has methodological and substantive errors and should not be published in its current form without taking my recommendations into account.

The temperature of 27oC is very high for aphids and inhibits their development and fecundity. I understand that the authors chose to use the temperature projected in the end of this century. However, I believe that there is a lack of research at a control temperature - optimal for the development of this species.

I do not negate the age-stage, two-sex life table used by the authors. I asked to compare the obtained results with another method - the authors did not do it. I believe that such a high Net reproductive rate Ro is wrong. The species would not achieve such high parameters, the more when it is exposed to stressful conditions such as high temperature and increased CO2 concentration.

The explanation of the authors that: "The age-stage, the two-sex life table was used in this study, primarily because of its ability to incorporate the male component of a population" is inadequate in the case of aphids. I'm afraid the authors don't know the biology of the aphids. Aphids develop parthenogenetically, giving birth to larvae throughout the growing season. The sexual generation appears only in the autumn, which is related to the shortening of the day and lowering the temperature. There are never 50% of the male in the aphid population. Even more so now, with global warming, there is a tendency to reduce the number of males in autumn. Therefore, the statement that the male, which are normally 50% of a population and do contribute significantly to its ecology, is incorrect in the context of aphids.

I encourage you to read the results of the work: Seo BY, Kim EY, Ahn JJ, Kim Y, Kang S, Jung JK. Development, Reproduction, and Life Table Parameters of the Foxglove Aphid, Aulacorthum solani Kaltenbach (Hemiptera: Aphididae), on Soybean at Constant Temperatures. Insects. 2020; 11 (5): 296. Published 2020 May 11.doi: 10.3390 / insects11050296.

For A. solani the authors compared traditional age-specific life table and the age – stage, two-sex life table developed by Chi and Liu and Chi. The obtained results regarding demographic parameters do not differ significantly.

The figures are still illegible and I am unable to properly read the data and interpret it properly.

This sentence is still not corrected - line 177-179: Significantly lower age-specific fecundity (mx) were found in the 1st and 3rd generations in 600 μL / L treatment compared with 400 μL/L and 600 μL / L CO2 treatments, which indicated that 600 μL / L CO2 treatment had the inhibitory effect on reproduction of L. erysimi (Fig. 3). How can you compare 600 μL / L treatment with 600 μL?

Verse 133-134 - the authors did not study Nilaparvata lugans. The research concerned L. erysimi.

Moreover, I still believe that the discussion is insufficiently carried out. In my opinion, adding a few sentences did not improve this section.

Author Response

Dear Chief Editor and Reviewer,

How are you! And Happy New Year!

Thank you very much for giving us the opportunity to revise our manuscript (ID insects-1497611-R1). We appreciate the reviewer very much for his/her constructive comments and suggestions on our manuscript. We have tried our best to revise our manuscript according to the reviews’ comments letter by letter, and the amendments are highlighted in red in the revised manuscript. The responses are below. Please check it!

Thanks you for your consideration to our manuscript. We hope this revised manuscript can meet the requirements of the reviewer and Insects. And we look forward to publishing our manuscript on Insets! If you have any question, please contact with me.

Thank you for your consideration.

Sincerely,

Dr. Gang Wu

Hubei Insect Resources Utilization and Sustainable Pest Management Key   Laboratory, College of Plant Science and Technology, Huazhong Agricultural University, Wuhan 430070, China. Tel / Fax: +86 27 87282130. E-mail: [email protected]

2022-1-2

Responses to the reviewer’s comments

Reviewer #1

Q1:.Thanks for the authors' answers. Some I can accept, but not all. The authors also did not follow all of my recommendations. I believe that the work sent for re-evaluation has methodological and substantive errors and should not be published in its current form without taking my recommendations into account.

Responses: Thank you very much for giving us the opportunity to revise our manuscript. In the revised manuscript, we have revised our manuscript according to and reviewer’ opinions. We have tried our best to revise our manuscript according to the reviews’ comments letter by letter, and the amendments are highlighted in red in the revised manuscript. We hope this revised manuscript can meet the requirements of the reviewer and Insects. And we look forward to publishing our manuscript on Insets! And we have lined out the right revised sentence by red words in the revised manuscript. Please check it.

Q2:.The temperature of 27oC is very high for aphids and inhibits their development and fecundity. I understand that the authors chose to use the temperature projected in the end of this century. However, I believe that there is a lack of research at a control temperature - optimal for the development of this species.

Responses: Yes. Indeed. I quiet agree with the reviewer’s opinions of the reviewer. The most suitable temperature for the growth and reproduction of aphids is 16~22 oC.

The present experiment is mainly to carry out the response of Lipaphis erysimi to extreme high temperatures (such as 27 oC), and clarify the impact of future climate warming on the growth and population outbreaks of L. erysimi. In our later experiments, we should carry out the influence of suitable temperature (i.g. 16~22 oC) on L. erysimi according to the reviewer’s suggestion, which can more truly reflect the response of L. erysimi. to environmental changes. Thanks again to the reviewer for his/her opinions

Q3:I do not negate the age-stage, two-sex life table used by the authors. I asked to compare the obtained results with another method - the authors did not do it. I believe that such a high Net reproductive rate Ro is wrong. The species would not achieve such high parameters, the more when it is exposed to stressful conditions such as high temperature and increased CO2concentration.

Responses: Responses: Yes. Indeed. The net reproductive rate (R0)was high in this manuscript! We had compared with the results among Fallahpour et al.(2015), Qayyum et al.(2018) and our experiment. The reason may be the impact factor, such as global warming factor in our experiment, and the impact factor of host plants and fertilization in Fallahpour et al. (2015) and Qayyum et al. (2018). The reviewer recommend the classic traditional life table by Watt and White (1977) and the statistical analysis methods by Seo et al. (2020). Thus, based on the traditional life table analysis method, Seo et al. (2020) used the age–stage, two-sex life table theory (Chi and Liu, 1985) and the method described by Chi (Chi, 1988) using the computer program TWOSEX-MSChart (Chi, 2020) to analysis the life history data of A. solani . The intrinsic rate of increase (r) is calculated using the Euler–Lotka formula (Goodman, 1982). These results do not prove that the traditional age-specific life table can be better to calculate the life table data of aphid. Based on two-sex life table theory, the Figure 4 in Seo et al study can be got, but traditional life table by Watt and White (1977) can be not.

Thus, Seo et al. (2020) and our study made graphs and analyzed the data using the age–stage, two-sex life table theory on the basis of perfecting the traditional life table. These two statistical analysis methods of two-sex life table (Chi and Liu, 1985; Chi, 1988; Chi, 2020) and traditional life table (Watt and White, 1977) have their own advantages. Thanks again to the reviewer for his/her opinions

Q4:The explanation of the authors that: "The age-stage, the two-sex life table was used in this study, primarily because of its ability to incorporate the male component of a population" is inadequate in the case of aphids. I'm afraid the authors don't know the biology of the aphids. Aphids develop parthenogenetically, giving birth to larvae throughout the growing season. The sexual generation appears only in the autumn, which is related to the shortening of the day and lowering the temperature. There are never 50% of the male in the aphid population. Even more so now, with global warming, there is a tendency to reduce the number of males in autumn. Therefore, the statement that the male, which are normally 50% of a population and do contribute significantly to its ecology, is incorrect in the context of aphids.

Responses: Thanks. I'm terribly sorry I didn't explain it clearly last time, which may made confused. I just want to explain why I used two-sex life table. One of advantages of two-sex life table is that it takes into male population when compared with traditional life table. However, the stage differentiation and the variable developmental rate among individuals is also ignored in traditional life table. For most insects though, developmental rates differ between the sexes and among individuals. Chi & Liu (1985) pointed out that neglecting the variable developmental rate and male population may cause errors in calculating demographic parameters, such as the intrinsic rate of increase, net reproductive rate and the mean generation time. Chi and Liu (1985) and Chi (1988) developed an two-sex life table to take the stage differentiation and the male population into consideration. Based on the age-stage, two-sex life table, Chi & Getz (1988) constructed a mass-rearing program for stage-structured populations. Furthermore, mathematical proofs demonstrating the correctness of applying the age-stage, two-sex life table to insect populations were provided by Yu et al. (2005) and Chi and Su (2006).

Although aphid develop parthenogenetically during the growing season, life table of aphid with parthenogenetic can also be analyzed based on two-sex life table theory, such as Seo et al. (2020) and Wang et al. (2019). So, I think our data from the two-sex life table can partially clarify the response of aphids to elevated CO2.

We hope this explanation can meet the requirements of the reviewer.

References was as follow:

Chi, H. Life-table analysis incorporating both sexes and variable development rates among individuals. Environ. Entomol.1988; 17, 26–34.

Chi, H, Liu, H. Two new methods for the study of insect population ecology. Bull. Inst. Zool. 1985; 24, 225–240.

Chi, H. and Getz, W.M. Mass rearing and harvesting based on an age-stage, two-sex life table: A potato tuber worm (Lepidoptera: Gelechiidae) case study. Environmental Entomology, 1988; 17, 18–25.

Yu, J.Z., Chi, H. and Chen, B.H. Life table and predation of Lemnia biplagiata (Coleoptera: Coccinellidae) fed on Aphis gossypii (Homoptera: Aphididae) with a proof on relationship among gross reproduction rate, net reproduction rate, and preadult survivorship. Annals of the Entomological Society of America, 2005; 98, 475–482.

Chi, H. and Su, H.Y. Age-stage, two-sex life tables of Aphidius gifuensis (Ashmead) (Hymenoptera: Braconidae) and its host Myzus persicae (Sulzer) (Homoptera: Aphididae) with mathematical proof of the relationship between female fecundity and the net reproductive rate. Environmental Entomology, 2006; 35, 10–21.

Wang, L., Wang, Q., Wang, Q., Rui, C., Cui, L. The feeding behavior and life history changes in imidacloprid‐resistant Aphis gossypii glover (Homoptera: aphididae). Pest Management Science, 2020; 76(4)). In our study, aphids are also parthenogenetic.

Q5:I encourage you to read the results of the work: Seo BY, Kim EY, Ahn JJ, Kim Y, Kang S, Jung JK. Development, Reproduction, and Life Table Parameters of the Foxglove Aphid, Aulacorthum solani Kaltenbach (Hemiptera: Aphididae), on Soybean at Constant Temperatures. Insects. 2020; 11 (5): 296. Published 2020 May 11.doi: 10.3390 / insects11050296.

For A. solani the authors compared traditional age-specific life table and the age – stage, two-sex life table developed by Chi and Liu and Chi. The obtained results regarding demographic parameters do not differ significantly.

Responses: Thanks. Although the demographic parameters between these two life table methods do not differ significantly, the data are not exactly same, especially for the intrinsic rate of increase (rm). For the finite rate of increase (λ), this value is generally reserved four decimal places. While in this study, just two decimal places were be reserved, so we didn’t see the difference in data from two method of life table. In addition, these results do not prove that the traditional age-specific life table can be better to calculate the life table data of aphid. Based on two-sex life table theory, the Figure 4 in Seo et al study can be got, but traditional life table can be not. Thanks again to the reviewer for his/her opinions

Q6:The figures are still illegible and I am unable to properly read the data and interpret it properly.

Responses: Thank you very much for your reminder. We have improved the resolution of the image and polished the Figures according to your suggestion. We hope this revised manuscript can meet your requirements and Insects.

Q7:This sentence is still not corrected - line 177-179: Significantly lower age-specific fecundity (mx) were found in the 1st and 3rd generations in 600 μL / L treatment compared with 400 μL/L and 600 μL / L CO2 treatments, which indicated that 600 μL / L CO2 treatment had the inhibitory effect on reproduction of L. erysimi (Fig. 3). How can you compare 600 μL / L treatment with 600 μL?

Responses: Thank the reviewer for your useful advice and the valuable documents! I'm SO SORRY thus mistake in our manuscript. We have checked and revised this mistake! Please check it in the line 176-177 of page 6.

Q9:Verse 133-134 - the authors did not study Nilaparvata lugans. The research concerned L. erysimi.

Responses: Thank you very much for your reminder. I'm terribly sorry for making such a mistake. We have checked and revised this mistake! Please check it in the line 131 of page 3.

Q10: Moreover, I still believe that the discussion is insufficiently carried out. In my opinion, adding a few sentences did not improve this section.

Responses: Thank the reviewer for his/her useful suggestion! In the revised manuscript, we had discussed our results and the life table with Yu et al. (2005), Chi and Su (2006), Seo et al. (2020) and Wang et al. (2019). Please check it in the line 249 -260 of page 16.
